# Physiological and Biochemical Responses of Almond (*Prunus dulcis*) Cultivars to Drought Stress in Semi-Arid Conditions in Iran

**DOI:** 10.3390/plants14050734

**Published:** 2025-02-27

**Authors:** Esmaeil Safavi Bakhtiari, Asghar Mousavi, Mehrab Yadegari, Bijan Haghighati, Pedro José Martínez-García

**Affiliations:** 1Faculty of Agriculture, Shahrekord Branch, Islamic Azad University, Shahrekord 8813733395, Iran; e.safavi7@gmail.com; 2Horticulture Crops Research Department, Chaharmahal and Bakhtiari Agricultural and Natural Resources Research and Education Center, Agricultural Research, Education and Extension Organization (AREEO), Shahrekord 8813657351, Iran; sa.mousavi@areeo.ac.ir; 3Research Center of Nutrition and Organic Products (R.C.N.O.P), Shahrekord Branch, Islamic Azad University, Shahrekord 8813733395, Iran; mehrabyadegari@gmail.com; 4Soil and Water Research Department, Chaharmahal and Bakhtiari Agricultural and Natural Resources Research and Education Center, Agricultural Research, Education and Extension Organization (AREEO), Shahrekord 8813657351, Iran; bhaghighati@yahoo.com; 5Department of Plant Breeding, Centre of Edaphology and Applied Biology of Segura, Spanish National Research Council (CEBAS-CSIC), 30100 Murcia, Spain

**Keywords:** fruit trees, antioxidant enzymes, water deficit, osmolytes, rootstocks

## Abstract

Identifying and selecting almond cultivars with drought tolerance traits is crucial for developing more resilient cultivars, especially in regions prone to water scarcity or facing changing climate conditions. In this study, the physiological and biochemical responses of different almond cultivars to water stress were evaluated using a randomized complete block design (RCBD) with three replications at the Agricultural and Natural Resources Research Center of Chaharmahal and Bakhtiari Province, Shahrekord, Iran, during the 2020 and 2021 growing seasons. During each season, the drought stress treatments were applied for four months prior to the collection of leaf tissue and assessment of the physiological and biochemical traits of the treated trees. In general, significant differences were observed for the different effects considered in the fitted model (years, repetitions, cultivars, drought treatments). The relative water content, as well as the chlorophyll a and b contents in the leaves of the evaluated cultivars, significantly decreased with increasing stress intensity. However, the total phenol content and the activities of antioxidant enzymes increased in response to drought stress. There were considerable differences in the studied cultivars’ responses to increasing drought intensity. According to the results, cultivars “Shahrood 8”, “Garnem”, and “Shahrood 12” demonstrated a high antioxidant capacity and the highest resistance, as observed through a smaller reduction in the relative water content under severe drought stress compared with the other cultivars. These results provide valuable insights that contribute to the development of more resilient almond cultivars and rootstocks, particularly in regions susceptible to water scarcity or those experiencing changing climatic conditions.

## 1. Introduction

Water scarcity in arid and semi-arid regions is the primary limiting factor affecting the yield and production of various horticultural crops, with the global reduction in water sources exacerbating challenges worldwide [1]. Coupled with high irradiance and elevated summer temperatures, water scarcity is affecting some of the main global agricultural regions [2,3]. Under the current climate-warming scenario, drought periods in these regions are expected to become more frequent [2], further straining agricultural systems. A plant’s response and ability to overcome water deficit or drought stress conditions are complex and determine the sensitivity or tolerance of plants exposed to water limitations [4]. In this sense, a series of mechanisms at the morphological, physiological, biochemical, cellular, and molecular levels have evolved in plants [5].

The *Prunus* genus comprises essential fruit crops, including almond (*Prunus dulcis*), a species widely distributed across diverse environmental conditions [6]. These environments usually have a complex terrain, a shortage of water resources, and high irrigation costs [6]. Clearly, drought profoundly impacts the morphology, physiology, and biochemistry of fruit trees, threatening crop productivity [7,8,9,10]. Although the almond tree is recognized as a drought-tolerant crop [11], water limitation in arid and semi-arid conditions restricts nut yield and quality [12,13]. To improve almond productivity, different water-saving irrigation methods, such as deficit irrigation (DI), for efficient water use can be applied without affecting the yield or quality [14,15]. These strategies can contribute to sustainable agriculture, providing greater benefits to farmers through the commercialization of these almonds as hydroSOS products [13,15]. However, while DI might be suitable for drought-tolerant cultivars, it is not recommended for water-sensitive trees due to the high risk of yield reduction [16].

To improve drought resistance in almond, the effects of drought stress on different almond genotypes have been extensively studied [17,18,19,20,21,22,23,24,25]. Additionally, native germplasms of almond, as important genetic sources, can be evaluated for their tolerance to various environmental stresses, especially drought stress [26,27]. A clear example is *P. ramonensis*, an almond species with small leaves and higher photosynthetic activity that is endemic to the hyper-arid Negev desert; it is seemingly unaffected by drought and has been well maintained under drought stress [28].

In many studies, various drought-related traits, including root traits [29], leaf traits [18,30], osmotic adjustment and turgor maintenance, leaf turgor and stomatal conductance [17,20,21,22,23], stem and leaf water potential [22,31,32], stomatal size and density [18], chlorophyll content [23,24], and chlorophyll fluorescence [23,32] have been assessed as possible indicators to evaluate the drought resistance of different genotypes and to improve our understanding of all of the mechanisms involved in almond’s response to drought stress.

Some of these studies proposed markers for assessing drought stress in almond, such as carotenoids, which act as chlorophyll protectors and are defense compounds with antioxidant roles that neutralize the harmful effects of reactive oxygen species under stress [32]. The root dry weight/leaf area ratio, a smaller stomatal size, and a lower specific leaf area (SLA), can potentially be good markers for drought tolerance [18]. In general, when conducting a short-term drought stress experiment in almond and other *Prunus* species, physiological parameters, such as the net photosynthesis rate (P_N_) and leaf water potential (ψ_w_), are good parameters to detect water deficits quickly; however, neither parameter can be measured in species with small leaves, such as *P. webbii* [23]. In contrast, SPAD and Fv/Fm are poor parameters for assessing drought response, as they seem to be unaffected or to have a late response in comparison to parameters such as P_N_ and ψ_w_ [23]. In addition, leaf greenness, leaf size, shoot growth, shoot dry weight, total leaf dry weight (TLDW), and stomatal density are poor markers for drought resistance in almond seedlings [18].

Despite the progress being made in understanding drought resistance and breeding in almond, the fact that this crop struggles to address water scarcity and future food demands still needs to be addressed. Identifying and selecting cultivars with enhanced drought tolerance traits can contribute to the development of more resilient almond cultivars, especially in regions exposed to water scarcity or facing changing climate conditions. This study aimed to identify tolerant cultivars under the semi-arid conditions in Iran by evaluating the physiological and biochemical responses to different drought stress intensities.

## 2. Results

### 2.1. The Effect of Drought Stress on Relative Water Content (RWC) and Chlorophyll Content in Almond Cultivars

Significant differences were identified in relative water content (RWC) between years, cultivars, and drought treatments and for the interaction drought stress × cultivar (Appendix A). Year 1 showed the highest RWC. Across all the almond cultivars, drought stress led to a significant reduction in RWC (Figure 1A). In stressed conditions, “Shahrood 7” exhibited the highest RWC in leaves, reaching 83.55%, while “Shahrood 8” had the highest mean values across drought treatments (73.15%). Under severe drought stress, the “Garnem” and “Shahrood 8” cultivars showed the smallest reduction in RWC (Figure 1A). Conversely, the “Mamaei”, “Shahrood 13” and “Shahrood 12” cultivars experienced the most pronounced reductions in RWC under severe drought stress conditions, from 72.61 to 45.86%, 72.71 to 45.86% and 73.49 to 55.16%, respectively, equating to reductions of 26.74%, 26.85% and 28.07% (Appendix A).

Regarding the chlorophyll a and b contents in almond leaves, the analysis of variance showed significant differences between years, repetitions (for chlorophyll b), drought treatments, and cultivars. For both traits, all interactions were non-significant. In both cases, the chlorophyll content in the second year was lower than that in the first year (Appendix A). The chlorophyll content experienced a reduction (non-significant) with drought treatments (Figure 1B,C). Among the cultivars, “Shahrood 8” and “Garnem” (8.46 and 8.19 mg g^−1^ fresh weight (FW), respectively) exhibited the highest levels of chlorophyll a (Appendix A), and “Garnem” (2.81 mg g^−1^ FW) had the highest content of chlorophyll b (Appendix A). Conversely, the lowest contents of both chlorophyll a (5.98 mg g^−1^ FW) and b (2.08 mg g^−1^ FW) were recorded in the “Saba” cultivar.

### 2.2. The Effect of Drought Stress on Malondialdehyde (MDA) Accumulation and Electrolyte Leakage (EL) in Almond Cultivars

An analysis of variance revealed significant differences in malondialdehyde (MDA) accumulation in almond leaves between years, drought treatments, and cultivars, and for the rep × year, drought treatment × year, and drought treatment × cultivar interactions (Appendix A). Significantly lower values were observed for MDA levels in the second year compared with the first year. A significant increase in MDA levels for all genotypes was observed with the increase in drought stress (Figure 2A and Appendix A). Under severe drought stress, the highest values were recorded for the “Shahrood 13” (46.65 µmol g^−1^ FW) and “Rabie” (45.10 µmol g^−1^ FW) cultivars, while the “Shahrood 8” (29.26 µmol g^−1^ FW), “Shahrood 10” (29.87 µmol g^−1^ FW), and “Garnem” (29.88 µmol g^−1^ FW) cultivars exhibited the lowest MDA accumulation levels (Appendix A).

A 3- to 4-fold increase in electrolyte leakage (EL) levels was also observed with increasing drought stress. Significant differences were observed between years, repetitions, drought treatments, cultivars, and for the rep × year, drought treatment × year, cultivar × year, and drought treatment × cultivar interactions. The second year showed lower EL levels than the first year. A clear, significant increase in EL levels was observed for all cultivars with the increase in drought stress (Figure 2B). Under drought stress, the “Garnem”, “Shahrood 12”, and “Shahrood 8” cultivars exhibited the lowest EL (37.84%, 38.22%, and 38.53%, respectively), while the “Shahrood 13” cultivar displayed the highest EL (61.22%) (Appendix A).

### 2.3. The Effect of Drought Stress on Proline Content and Soluble Carbohydrate Content in Almond Cultivars

The analysis of variance revealed significant differences in the proline content in almond leaves between years, repetitions, drought treatments, cultivars, and the repetition × year, drought treatment × year, and drought treatment × cultivar interactions (Appendix A). Proline content was significantly lower in year 1. A significant increase in the proline content with increasing drought intensity was observed for all cultivars (Figure 3A). Under non-stress conditions, the “Shahrood 6” and “Saba” cultivars displayed the highest (23.38 mg g^−1^ FW) and lowest (15.11 mg g^−1^ FW) proline contents, respectively. Under severe drought stress, the “Shahrood 6” cultivar exhibited the highest proline content (60.99 mg g^−1^ FW), while the “Rabie” cultivar had the lowest proline content (41.27 mg g^−1^ FW) (Appendix A).

Regarding the soluble carbohydrate content, the analysis of variance showed significant differences between years, drought treatments, cultivars, and the rep × year, drought × year, and drought × cultivar interactions. A significant increase in the soluble carbohydrate content was observed for all cultivars with increasing drought intensity (Figure 3B). The measured carbohydrate content in the second year was significantly higher than in the first year. Under non-stress conditions, the “Garnem” and “Shahrood 12” cultivars (68.47 and 63.96 mg g^−1^ FW, respectively) displayed the highest values, while “Araz” (47.82 mg g^−1^ FW) had the lowest soluble carbohydrate content. Under severe drought stress, the highest contents of soluble carbohydrates were found in the “Garnem” (105.16 mg g^−1^ FW), “Shahrood 8” (104.14 mg g^−1^ FW), and “Shahrood 12” (103.03 mg g^−1^ FW) cultivars. Conversely, the “Araz”, “Shahrood 6”, and “Saba” cultivars had the lowest soluble carbohydrate contents under stress conditions (Appendix A).

### 2.4. The Effect of Drought Stress on Total Phenol Content in Almond Cultivars

Significant differences in the phenol content in almond leaves were observed between years, repetitions, drought treatments, cultivars, and the repetition × year and drought treatment × cultivar interactions. Remarkably, the total phenol content in the second year was significantly higher than that in the first year. Across all the almond cultivars, drought stress induced a significant increase in the phenol content, with the highest content being recorded under severe drought stress conditions (Figure 4). Under non-stress conditions, the “Shahrood 12” (18.07 mg GAE/g FW), “Garnem” (18.10 mg GAE/g FW), and “Shahrood 8” (18.67 mg GAE/g FW) cultivars displayed the highest total phenol contents, while the “Saba” cultivar exhibited the lowest amount of total phenol (12.46 mg GAE/g FW). Specifically, under severe drought stress, the “Garnem” (41.05 mg GAE/g FW), “Shahrood 8” (43.32 mg GAE/g FW), “Mamaei” (43.86 mg GAE/g FW), and “Shahrood 12” (45.53 mg GAE/g FW) cultivars exhibited the highest phenol contents (Appendix A).

### 2.5. The Effect of Drought Stress on Antioxidant Enzyme Activities in Almond Cultivars

Regarding catalase (CAT) enzyme activity, the analysis of variance showed significant differences between years, repetitions, drought treatments, cultivars, and the repetition × year, drought treatment × year, and drought treatment × cultivar interactions. CAT activity measured in the second year was significantly higher than that in the first year. CAT activity increased significantly with the intensity of drought stress in all of the cultivars (Figure 5A). Significant differences in CAT activity were observed among almond cultivars under various levels of drought stress; the “Shahrood 12” (26.08 U mg^−1^ protein) and “Garnem” (25.18 U mg^−1^ protein) cultivars showed the highest CAT activity, while the “Mamaei” (7.65 U mg^−1^ protein) cultivar had the lowest CAT activity (Appendix A, Figure 5A).

The year, drought treatment, and the interaction drought treatment × cultivar had significant effects on peroxidase (POX) enzyme activity. Particularly, the POX activity during the second year was higher than that in the first year. Severe drought stress caused a 32.03% increase in POX activity compared to non-stress conditions. Among cultivars, “Garnem” showed the highest POX activity (15.24 U mg^−1^ protein) under severe drought stress, while the lowest activity was observed in “Rabie” (8.54 U mg^−1^ protein), “Shahrood 7” (9.24 U mg^−1^ protein), and “Araz” (9.66 U mg^−1^ protein) (Figure 5B).

Drought treatment, cultivar, and the year × drought and drought × cultivar interactions had significant effects on superoxide dismutase (SOD) enzyme activity (Figure 5C). SOD activity in the second growing season was lower than that in the first. Drought stress induced a significant increase in SOD activity across all almond cultivars, with the highest activity being observed under severe drought stress. Under non-stress conditions, the “Shahrood 12” (20.65 U mg^−1^ protein) and “Saba” (10.92 U mg^−1^ protein) cultivars exhibited the highest and lowest SOD activity, respectively (Appendix A). Conversely, under severe drought stress, the highest SOD activity was found in the “Garnem” cultivar (38.76 U mg^−1^ protein), while the lowest activities were recorded in the “Saba” (18.77 U mg^−1^ protein), “Shahrood 13” (20.23 U mg^−1^ protein), “Shahrood 6” (20.32 U mg^−1^ protein), and “Shahrood 21” cultivars (20.92 U mg^−1^ protein) (Figure 5C).

According to the analysis of variance, glutathione peroxidase (GPX) enzyme activity was significantly affected by year, drought stress treatment, and cultivar, and the year × drought and drought × cultivar interactions (Figure 5D). GPX activity recorded in the second studied year exceeded that of the first. Under non-stress conditions, the “Rabie” (8.45 U mg^−1^ protein), “Shahrood 21” (8.70 U mg^−1^ protein), “Shahrood 13” (9.51 U mg^−1^ protein), and “Mamaei” (9.29 U mg^−1^ protein) cultivars had the lowest GPX activity, while the “Garnem” (19.00 U mg^−1^ protein), “Shahrood 8” (18.89 U mg^−1^ protein), and “Shahrood 12” (18.74 U mg^−1^ protein) cultivars exhibited the highest activity (Figure 5D). Under severe drought stress, the highest GPX activities were observed in the “Shahrood 8” (37.16 U mg^−1^ protein) and “Garnem” (37 U mg^−1^ protein) cultivars, while the lowest activities were recorded in the “Rabie” (15.12 U mg^−1^ protein) and “Shahrood 13” cultivars (15.82 U mg^−1^ protein) (Figure 5D).

## 3. Discussion

We conducted a comprehensive study on the physiological and biochemical responses of 13 almond cultivars and 1 hybrid rootstock under different levels of drought stress over two growing seasons. Understanding their suitability is crucial for selecting genotypes for almond breeding programs aimed at enhancing the drought tolerance of new cultivars and rootstocks.

A reduction in available water intensifies osmotic pressure, resulting in a diminished amount of water being accessible to the plant. This reduced water absorption under water deficit conditions led to a lower relative water content (RWC) in the leaves compared to that under non-stress conditions (Figure 1A). Alvarez et al. [29] previously reported drastic decreases in the relative leaf water content and osmotic potential in almond trees with increasing drought intensity. Similarly, Prgomet et al. [21] showed that the relative water content was always significantly lower in non-irrigated almond trees compared to all other treatments. In almond, non-significant changes in RWC were associated with more tolerance to water stress [18].

A progressive decrease in the leaves’ relative water content (RWC) reduces stomatal conductance (gs), which slows CO_2_ assimilation (A) until it halts, leading to CO_2_ release and consequently reducing leaf photosynthesis rates and chlorophyll fluorescence parameters [33,34]. The gradual reduction in gs is attributed to a decrease in leaf water potential (ψ_w_) [22]. At the leaf level, the same gradual reduction in gs results in stomatal closure, which is the main response to water deficits in almond [35,36,37,38]. In our study, different trends in the decrease in RWC were observed among the different cultivars (the “Garnem” and “Shahrood 8” cultivars showed the lowest reduction in RWC, and “Mamaei”, “Shahrood 13”, and “Rabie” the highest), which likely resulted in different degrees of stomatal closure among the almond cultivars in response to water deficits, which has also been observed previously [17,18,39]. In this sense, higher photosynthetic activity, stomatal conductance, water use efficiency, embolism resistance, and stem growth were observed in *P. ramonensis* (a wild species of the hyper-arid Negev desert) under drought compared to the evaluated rootstock, confirming higher drought resistance in wild vs. cultivated species of almonds [28].

In the current study, as the drought levels increased, both electrolyte leakage (EL) and malondialdehyde (MDA) exhibited an upward trend, with variations depending on the cultivar (Figure 2A,B). Drought stress disrupts the electron transfer process in mitochondria and chloroplasts, triggering the generation of active oxygen radicals, which, in turn, leads to oxidative damage to cell membranes [40]. At the same time, symptoms of oxidative damage (such as lipid peroxidation) have been used to evaluate the increase in reactive oxygen species (ROS) production under drought stress [41]. In other trees, such as olive, significant increases in MDA levels of more than three times those of irrigated plants were observed in the leaves at the maximum level of drought stress [42]. Clearly, the observed increase in MDA levels seems to be a good indicator of damage in almond tree membranes and could be utilized in almond as a key indicator of this damage and of lipid peroxidation [43,44].

In our study, the contents of chlorophyll a (Figure 1B) and b (Figure 1C) in the leaves of the almond cultivars exhibited a non-significant decrease with increasing drought stress levels. In addition, the chlorophyll content seems to be unaffected or to have a late response to short-term drought in *Prunus* species [23]. In this sense, and as stated by [4], not all plants show a reduced chlorophyll content under drought stress.

Several studies have shown that different amino acids and/or carbohydrates—proline, γ-aminobutyric acid, sucrose, hexoses, maltose, and others—accumulate within cells in response to drought stress, which contributes to the regulation of osmotic potential in the plant by raising the osmotic pressure and prompting water to flow from vacuoles to the cytoplasm to regulate pressure and maintain turgor [8,21,45]. As was expected, this physiological mechanism typical of the synthesis and accumulation of different amino acids and/or carbohydrates was observed in this study. An incremental increase in the soluble proline and carbohydrate contents was observed in all the studied almond cultivars. However, there were significant differences in the level of this increase among cultivars. As has been suggested, this higher proline content could play an essential role in plant recovery after severe and moderate stresses [46].

Phenolic compounds, known for their potent antioxidant properties, are upregulated under drought, neutralizing free radicals and mitigating oxidative damage [47]. As suggested by Lipan et al. [13], the excess CO_2_ that accumulates under stress conditions is used for the biosynthesis of carbon-based secondary metabolites, resulting in an increase in total phenolic compounds (TPCs). A correlation between TPCs and the stress integral was previously observed in almond [17]. In this study, the observed increase in total phenol content in different almond cultivars (Figure 4) reflects a natural response that enhances the plant’s defense system against drought stress.

Clearly, drought stress leads to cellular oxidative stress due to the accumulation of ROS, with their concentration being tightly regulated by an efficient antioxidative system comprising enzymatic and non-enzymatic antioxidants. The activities of the enzymatic antioxidants catalase (CAT) and peroxidase (POX) increased under water deficiency conditions in different woody species, namely peach [48], grapevine [49], apple [50], and olive [51]. Additionally, Wang et al. [50] observed that the most drought-tolerant apple rootstock exhibited the highest antioxidant capacity relative to apple rootstocks with a low tolerance to drought. Furthermore, a transgenic plum line with improved antioxidant capacity seemed to confirm that precise regulation of CAT activity is associated with water stress tolerance [52]. In almond, the enzymatic activity of antioxidant enzymes is positively related to the photosynthetic state of the leaf and negatively related to the transpiration rate (EVAP), stomatal conductance (g_S_), net photosynthesis rate (P_N_), and leaf water potential (ψ_w_), which allows the plant to avoid dehydration [53]. The overall trend in the studied almond cultivars was an increase in antioxidant enzyme activity, including higher activities for CAT (Figure 5A), POX (Figure 5B), superoxide dismutase (SOD, Figure 5C), and glutathione peroxidase (GPX, Figure 5D) with stress intensity, each showing a varying degree of response. In a previous study in almond [53], both CAT and POX activities decreased during drought stress in almost all the species, except in *P. webbi* and *P. eleagnifolia*, which displayed high POX and constant CAT activities; of note, both of these species showed higher tolerance to drought stress based on various physiological parameters relative to the other observed species [23]. The high levels of these enzymes observed for some cultivars evaluated in this study may confirm a high tolerance to drought stress, similar to the drought-tolerant species described by Martínez-García et al. [23].

A distinct response to drought stress was evident across all cultivars, which was characterized by elevated levels of osmolyte compounds such as proline and soluble carbohydrates, alongside an enhanced antioxidant capacity involving phenolic compounds and antioxidant enzymes (CAT, POX, SOD, and GPX). Certain cultivars, such as “Garnem”, “Shahrood 8”, and “Shahrood 12”, exhibited greater tolerance to drought, manifesting as minimal reductions in relative water content (RWC) and high antioxidant activities under stress conditions. This is consistent with the study by Bielsa et al. [54], where the drought tolerance potential of “Garnem” was previously highlighted. The response of these three cultivars suggests that these specific cultivars possess robust mechanisms for maintaining a water balance and mitigating oxidative stress induced by drought and could indeed be valuable genetic resources for future almond breeding programs.

## 4. Materials and Methods

The experiments were conducted at the Agricultural and Natural Resources Research Center in Shahrekord, Iran (53°17′ N, 56°55′ E; altitude 2073 m), during the 2020 and 2021 growing seasons. Climatic data for the observed years are detailed in Table 1. The physicochemical characteristics of the soil at the experimental site are outlined in Table 2, including a medium and loamy soil texture. This study employed a split-plot design within a randomized complete block (RCBD) framework, including three replications. Within each replication, four trees were carefully selected for the assessment of physiological and biochemical traits.

This experiment primarily focused on implementing various irrigation regimes using the Time–Domain Reflectometer (TDR; Time-FM) method at four distinct levels (treatments): 70%, 50%, 30%, and 10% of moisture evaporation based on Field Capacity (FC). The treatments were no stress (70% FC), mild drought stress (50% FC), medium drought stress (30% FC), and severe drought (10% FC) stress. The sub-factor involved evaluating 13 commercial almond cultivars (“Mamaei”, “Rabie”, “Saba”, “Araz”, “Eskandar”, “Aidin”, “Shahrood 6”, “Shahrood 7”, “Shahrood 8”, “Shahrood 10”, “Shahrood 12”, “Shahrood 13”, and “Shahrood 21”) grafted onto the hybrid peach x almond clonal rootstock “GN 15” (Garnem). Additionally, one clonal rootstock, “Garnem”, was incorporated as a sub-factor, serving as a control without grafting. The experiment utilized two-year-old grafted plants, ensuring uniformity in terms of age, stem diameter, and height. After soil preparation, the three replications of the young grafted almond cultivars were planted in the field following the designated block design. Four experimental plots were set up for drought treatments, and then, each prepared plot was divided into fourteen sub-plots to investigate different cultivars’ responses to imposed drought stress. To induce drought stress, hygrometer tubes (TDR) were strategically installed in each experimental plot, determining the irrigation cycle based on the soil moisture content for different treatments. To determine the timing of irrigation for different levels of water stress, TDR readings were taken both before and after irrigation. Various drought levels were applied based on the amount of available moisture, expressed as a percentage of the moisture between FC and permanent wilting point (PWP), in the different treatments. The water source used for irrigation had an electric conductivity (EC) of 1.01 ds/m and pH 7.6. Drought stress was applied over a four-month period, commencing in early June, during both years. All essential cultivation practices (such as pruning, weed management or fertilization) were consistently applied to all treatment plots throughout the experimental period. Towards the end of the drought stress period, leaves from the treated trees for each cultivar were promptly sampled and immediately transferred to the laboratory for physiological and biochemical assessments.

### 4.1. Estimation of Relative Water Content (RWC) and Chlorophyll Content

The relative water content (RWC (%)) of the leaves was calculated using the formula (FW − DW)/(TW − DW) × 100, where FW represents the fresh weight of the leaves, DW is the dry weight recorded after drying the samples at 75 °C for 48 h, and TW is the turgor weight (determined by subjecting leaves to rehydration in distilled water for 24 h at room temperature), as outlined by [55].

Chlorophylls were extracted from leaves using 99% methanol. Following centrifugation at 12,000× *g* for 5 min, the absorbance of the solution was measured using a spectrophotometer (Unico, UV-2100, United Products & Instruments, Inc, Dayton, OH, USA) at 470, 653, and 666 nm. The chlorophyll a and b contents were estimated by employing the equations proposed by [56] and are expressed as milligrams of each pigment per gram of leaf FW.

### 4.2. Estimation of Malondialdehyde and Electrolyte Leakage

Lipid peroxidation levels were assessed by quantifying the decomposition product of peroxidized membrane polyunsaturated fatty acids, specifically malondialdehyde (MDA), as outlined by Davey et al. [57]. Fresh leaf samples (0.3 g) were ground in 3 mL of trichloroacetic acid (TCA, 0.1% *w*/*v*) and then centrifuged at 10,000× *g* for 20 min at 4 °C. Subsequently, 0.5 mL of the supernatant was mixed with 300 µL of thiobarbituric acid (TBA, 0.5% *w*/*v*) and 20% TCA (*w*/*v*). The samples underwent heating at 90 °C for 30 min. Following this incubation, the reaction was halted in an ice bath, and the absorbance of the supernatant was measured at 532 and 600 nm. The MDA concentration was determined using the specific extinction coefficient of 155 mM^−1^cm^−1^, and the results are expressed as µmol g^−1^ FW. Electrolyte leakage (EL) was employed to assess the stability of the leaf cell membrane by following the method outlined by [58]. In this approach, 2 cm leaf pieces are prepared, washed, and placed in vials with 10 mL of distilled water for 24 h in darkness. The initial leakage rate (EC1) was measured using a conductivity meter. Subsequently, the vials were autoclaved at 121 °C for 20 min to disrupt the leaf cells. After cooling, the secondary leakage rate (EC2) was recorded. EL was calculated using the following equation:EL (%) = (EC1/EC2) × 100

### 4.3. Estimation of Proline and Soluble Carbohydrate Contents

The proline content in the leaf samples was determined following the method described by Bates et al. [59]. Fresh leaf samples were homogenized in 5 mL of 3% sulfosalicylic acid, and 1 mL of this homogenate was added to acid-acetic ninhydrin reagent and glacial acetic acid. Subsequently, it was incubated at 100 °C for 1 h, and then cooled and extracted with toluene. The absorbance of the extraction was measured at 520 nm, and the proline content of each sample was obtained by applying an external calibration curve generated using L-proline and is reported in milligrams per gram of sample FW.

To measure the soluble carbohydrate content, 0.5 g of fresh leaves was homogenized with 5 mL of 99% methanol. The supernatant (0.2 mL) was then mixed with 3 mL of anthrone solution (150 mg of anthrone in 100 mL of 72% sulfuric acid, *w*/*w*). The samples underwent a 10 min boiling water bath. After cooling, the absorbance was measured at 625 nm [60]. The content of soluble sugars was determined using an external calibration curve based on glucose and is expressed as milligrams per gram of leaf FW.

### 4.4. Estimation of Total Phenol Content

The total phenol content was determined using Folin–Ciocalteu reagent following the method outlined by Wijeratne et al. [61] with slight modifications. Briefly, the leaf samples were homogenized and extracted with 99% methanol at room temperature. The samples were vortexed for 30 s and centrifuged at 20 °C for 10 min at 5000 rpm. After recovering the supernatant, methanol was added to the remaining extract, and the mixture was centrifuged again. Finally, the supernatant (200 μL) was diluted with 300 μL of distilled water, and subsequently, 2.5 mL of freshly prepared 50% Folin–Ciocalteu reagent was added. The solution was incubated at room temperature in the dark for 3 min. Following this incubation, 2 mL of 7.5% sodium carbonate solution was introduced. The absorbance of the solution was measured at 765 nm. The total phenol content was obtained using an external calibration curve based on gallic acid and is expressed as milligrams of gallic acid equivalent per gram of leaf FW.

### 4.5. Estimation of Antioxidant Enzyme Activity

For the enzyme assays, 0.5 g of each sample was homogenized in 5 mL of 50 mM potassium phosphate buffer (pH = 6.8) containing 1% PVP (polyvinylpyrrolidone) and 1 mM EDTA. After homogenization, the samples were centrifuged at 12,000× *g* for 20 min at 4 °C, and the resulting supernatant fraction was utilized as the source of both protein and enzymes (enzyme extract). The concentration of soluble protein in the extract was determined using the Bradford protein assay [62], with bovine serum albumin (BSA) serving as the standard.

Catalase (CAT) activity was assessed using a modified method based on [63]. CAT activity was estimated in millimoles of decomposed H_2_O_2_ per minute (one unit) per milligram of soluble protein, considering the H_2_O_2_ extinction coefficient of 39.4 mM^−1^ cm^−1^.

Peroxidase (POX) activity was measured using guaiacol as a substrate. The reaction mixture included potassium phosphate buffer (50 mM, pH = 7), H_2_O_2_ (1%), guaiacol (2%), and the enzyme extract. The absorbance increment resulting from guaiacol oxidation was determined using a spectrophotometer for continuous monitoring at 470 nm for 3 min [64]. POX activity is expressed as millimoles of tetraguaiacol produced per minute per milligram of soluble protein (U mg^−1^ protein) using the tetraguaiacol extinction coefficient of 25.5 mM^−1^ cm^−1^.

Superoxide dismutase (SOD) enzyme activity was assessed based on its ability to inhibit the photochemical reduction of nitro blue tetrazolium (NBT) by following the method described by [65]. The activity of SOD is reported as units per milligram of soluble protein.

The reaction solution used to assess glutathione peroxidase (GPX) activity consisted of 200 mM sodium phosphate buffer (pH 7.2), 1 mM EDTA, 1 mM NaN_3_, 0.15 mM NADPH, and 1 unit of glutathione reductase mixed with 100 µL of the enzyme extract. The reaction was initiated by adding 0.5 mM H_2_O_2_, and absorbance at 340 nm was monitored for 1 min, where a reduction in absorbance is attributed to the oxidation of NADPH to NADP^+^ [66]. GPX activity was calculated as millimoles of NADPH oxidized per minute per milligram of protein (units per mg of soluble protein) using an extinction coefficient of 6.62 mM^−1^ cm^−1^.

### 4.6. Statistical Analysis

The full statistical model used for the split-plot RCBD is as follows:$$Y_{ijkl}=\mu +\alpha_{i}+\varphi_{l}+\gamma_{k}+\theta_{ikl\;}+\beta_{j}+{\left(\alpha \beta \right)}_{ijl}+\varepsilon_{ijkl}$$
where γ is the effect of the $k^{th}$ block (or rep), α is the effect of the $i^{th}$ drought treatment (main plot), φ is the effect of the ${\;l}^{th}$ year, β is the effect of the $j^{th}$ genotype (sub-plot), and αβ is the interaction between the $i^{th}$ drought treatment and $j^{th}$ genotype. The main-plot error θ is uniquely identified by the (k) block (or rep) to which it belongs, by the drought treatment (i) with which it was treated, and the year (l) analyzed. Finally, $\varepsilon_{ijkl}$ is the residual random error term (sub-plot error). Both θ and $\varepsilon_{ijkl}$ are assumed to be normally distributed ${N(0,\;\sigma }^{2})$. A letter display based on Fisher’s LSD multiple comparison test was used when significant effects were found via the F-test, considering a significance level of *p* ≤ 0.05. The models for all of the traits were fitted using the R package stats [67]. All figures were created using the R package ggplot2 [68].

## 5. Conclusions

A distinct response to drought stress was evident across all cultivars, characterized by elevated levels of osmolyte compounds such as proline and soluble carbohydrates, alongside enhanced antioxidant capacities involving phenolic compounds and antioxidant enzymes (CAT, POX, SOD, and GPX). Certain cultivars, such as “Garnem”, “Shahrood 8”, and “Shahrood 12”, exhibited a greater tolerance to drought, as shown by minimal reductions in the relative water content (RWC), a high antioxidant activity, and elevated numbers of phenolic compounds under stress conditions. Future research should focus on understanding the genetic basis of drought resistance in the identified cultivars to develop robust molecular markers, accelerating breeding programs for enhanced resilience and ultimately contributing to the development of more climate-resilient almond cultivars.

## Figures and Tables

**Figure 1 plants-14-00734-f001:**
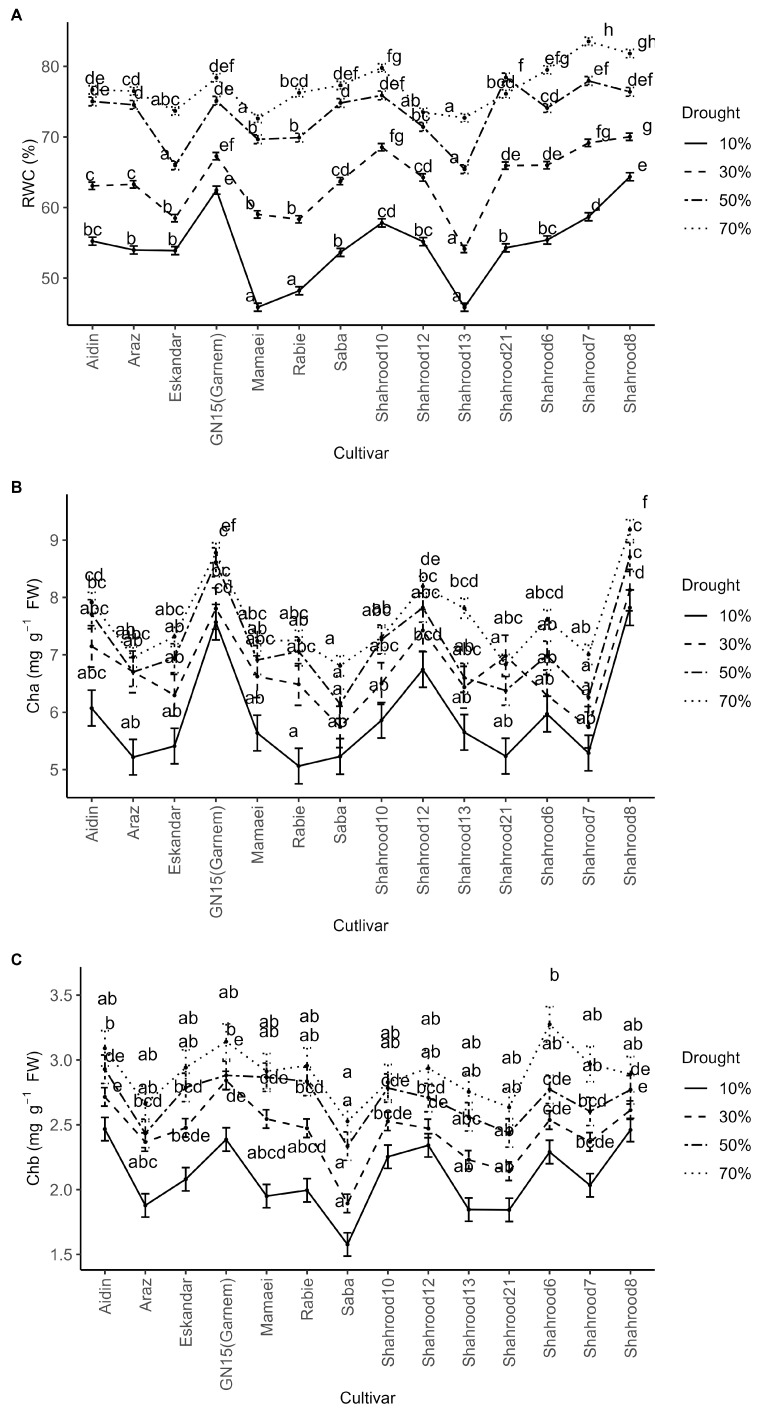
Effect of drought treatments on (**A**) relative water content (RWC (%)) (**B**) chlorophyll a (Cha) content (mg g^−1^ FW) and (**C**) chlorophyll b (Chb) content (mg g^−1^ FW) in the leaves of different almond cultivars. Error bars represent (estimated marginal) means ± standard error. Means with different letters are significantly different at the 5% level of significance, as determined by Tukey’s test.

**Figure 2 plants-14-00734-f002:**
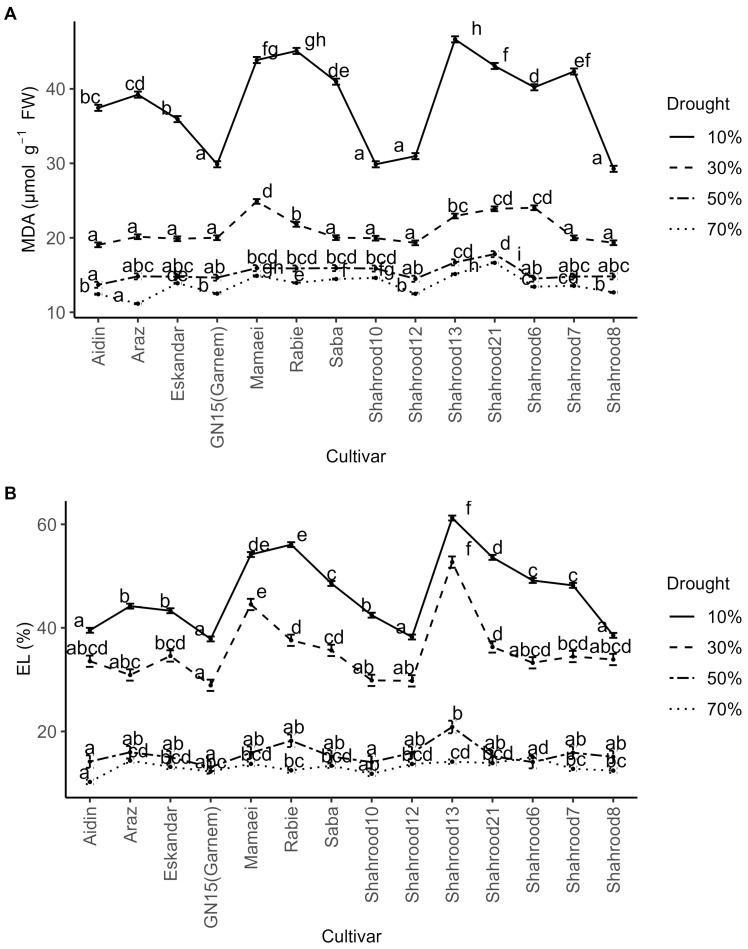
Effect of drought treatments on (**A**) malondialdehyde content (MDA) (µmol g^−1^ FW) and (**B**) electrolyte leakage (EL (%)) in the leaves of different almond cultivars. Error bars represent (estimated marginal) means ± standard error. Means with different letters are significantly different at the 5% level of significance, as determined via Tukey’s test.

**Figure 3 plants-14-00734-f003:**
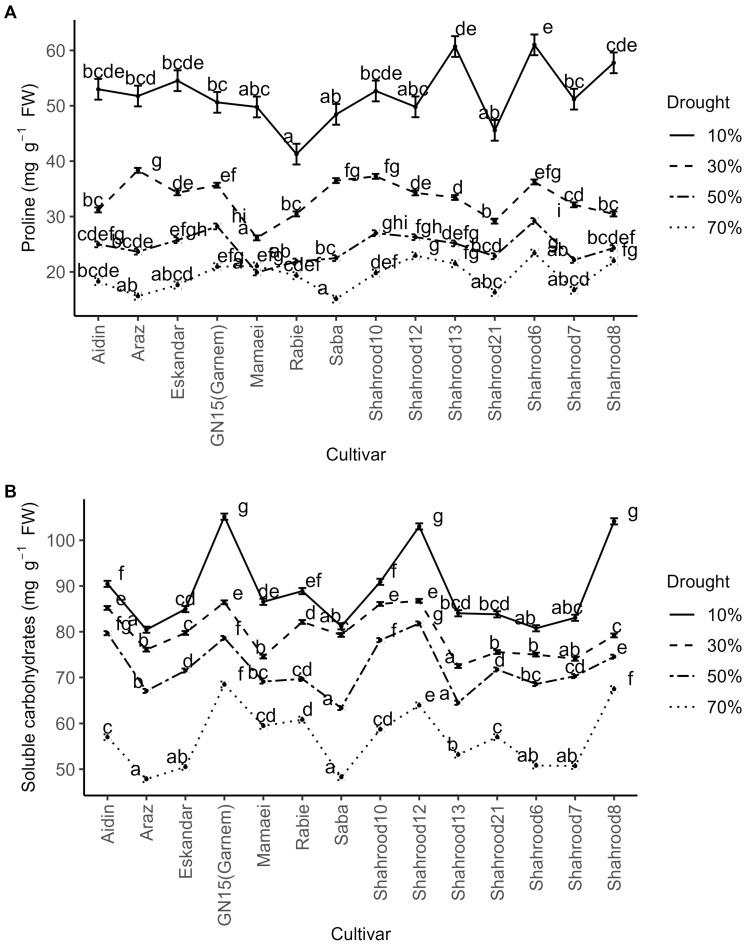
Effect of drought treatments on (**A**) proline content (mg g^−1^ FW) and (**B**) soluble carbohydrates (mg g^−1^ FW) in the leaves of different almond cultivars. Error bars represent (estimated marginal) means ± standard error. Means with different letters are significantly different at the 5% level of significance, as determined via Tukey’s test.

**Figure 4 plants-14-00734-f004:**
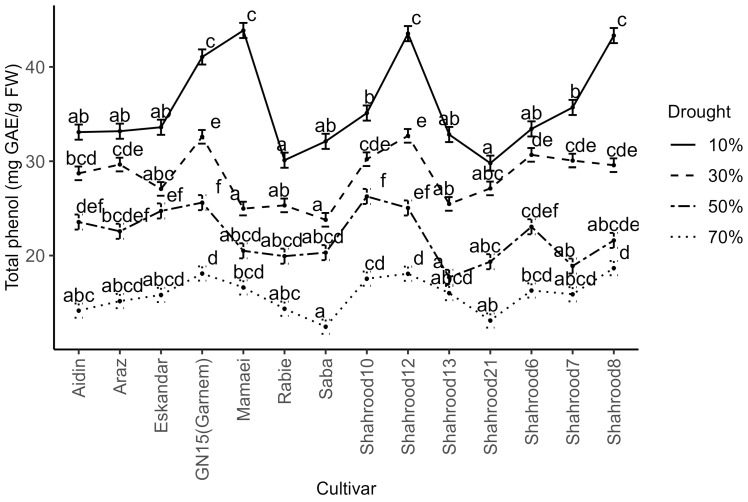
Effect of drought treatments on total phenol content (mg GAE/g FW) in the leaves of different almond cultivars. Error bars represent (estimated marginal) means ± standard error. Means with different letters are significantly different at the 5% level of significance, as determined via Tukey’s test.

**Figure 5 plants-14-00734-f005:**
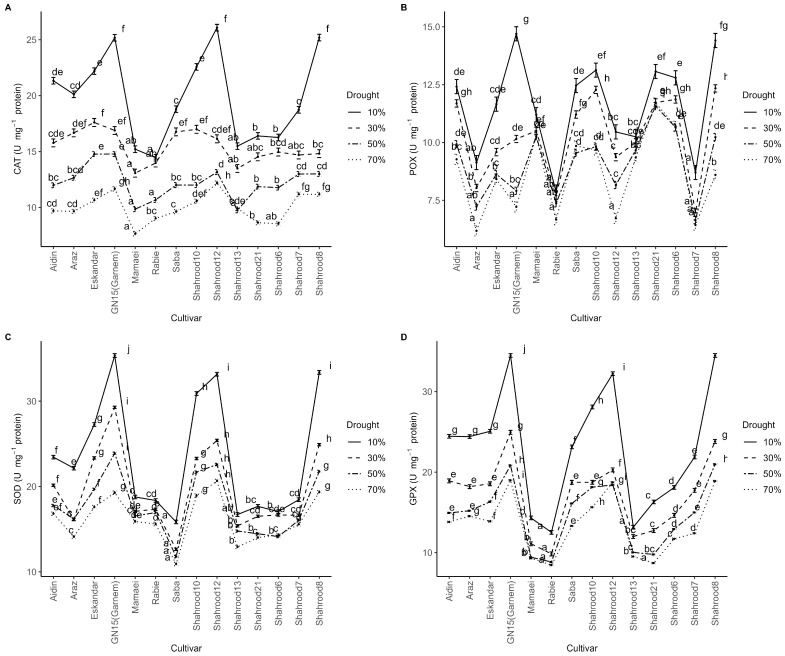
Effect of drought treatments on antioxidant enzyme activities in the leaves of different almond cultivars ((**A**): catalase (CAT) (U mg^−1^ protein); (**B**): peroxidase (POX) (U mg^−1^ protein); (**C**): superoxide dismutase (SOD) (U mg^−1^ protein); (**D**): glutathione peroxidase (GPX) (U mg^−1^ protein)). Error bars represent (estimated marginal) means ± standard error. Means with different letters are significantly different at the 5% level of significance, as determined via Tukey’s test.

**Table 1 plants-14-00734-t001:** Climatic conditions at the experimental site during the studied growing seasons.

Year	Total Precipitation (mm)	Average Humidity (%)	Average Wind Speed (m s^−1^)	Average Temperature (°C)	Maximum Temperature (°C)	Minimum Temperature (°C)
2020	259.5	38.0	2.8	12.7	37.2	−12.2
2021	198.9	35.6	3.1	13.5	38.6	−19.8

**Table 2 plants-14-00734-t002:** The physicochemical properties of the soil at the experimental site during the growing seasons studied.

Year	Texture	pH	Ec	O.C	N	P	K	Fe	Mn	Zn
		-	(dS m^−1^)	(%)		(mg kg^−1^)				
2020	Loamy	7.88	1.01	0.76	0.06	21.20	354.00	4.25	11.47	0.95
2021	Loamy	7.90	0.93	0.41	0.04	8.40	231.00	4.01	9.63	0.63

## Data Availability

Data is contained within the article or Appendix A.

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
