# Peer review of "Physiological and Biochemical Responses of Almond (*Prunus dulcis*) Cultivars to Drought Stress in Semi-Arid Conditions in Iran"

_plants, 2025, doi:10.3390/plants14050734_

Round 1

Reviewer 1 Report (Previous Reviewer 2)

Comments and Suggestions for Authors

The authors' revised paper deals with the effect of growing conditions on the phytochemical composition of 14 almond varieties. The impact of drought over two growing seasons in 2021 and 2022 was studied in detail.

Water and chlorophyll content, electrolyte levels, malonoaldehyde content, prolinephenol content (as the gallic acid equivalent), and antioxidant enzyme levels (catalase, glutathione peroxidase, superoxide dismutase) were studied.

The presentation of research results in the paper could be made more clear. Also, the graphs included in the supplementary materials should be clearer.

In the description of the enzyme studies, there are abbreviations (FW, EL) that have not been explained before. They should be explained the first time they are used in the text of the paper (lines 135, 136).

In line 436 insted of “extraction” the word “extract” should be used.

It should also be clarified in the preparation of the extract for polyphenol sum studies whether centrifugation took place with the plant material or whether it was previously filtered out.

The most serious objection to the content of the paper is the lack of a summary in the form of conclusions of the study. This section should be added to the paper.

Author Response

Dear Editor and Reviewers,

The author would like to gratefully acknowledge and sincerely thank the reviewers for their valuable comments. We have responded to all comments and have revised the paper in light of them. Detail of our responses to each comment are shown below. Comments, when requested, are shown in red.

We would like to comment that a complete English revision has been performed by the MDPI English service. In addition, new figures were created and all Annex were removed.

Reviewer 3)

The authors' revised paper deals with the effect of growing conditions on the phytochemical composition of 14 almond varieties. The impact of drought over two growing seasons in 2021 and 2022 was studied in detail.

Water and chlorophyll content, electrolyte levels, malonoaldehyde content, prolinephenol content (as the gallic acid equivalent), and antioxidant enzyme levels (catalase, glutathione peroxidase, superoxide dismutase) were studied.

The presentation of research results in the paper could be made more clear. Also, the graphs included in the supplementary materials should be clearer.

            Response: We have improved the text accordingly and new figures provided.

In the description of the enzyme studies, there are abbreviations (FW, EL) that have not been explained before. They should be explained the first time they are used in the text of the paper (lines 135, 136).

            Response: We have improved the text accordingly

In line 436 instead of “extraction” the word “extract” should be used.

            Response: We have improved the text: “The absorbance extracted was”…

It should also be clarified in the preparation of the extract for polyphenol sum studies whether centrifugation took place with the plant material or whether it was previously filtered out.

            Response: Thank you very much for the comment. We have updated the text accordingly. The extraction was performed by following the protocol describe by Wijerante et al. (2006) method with slight modifications. Fresh leaf tissue were homogenized and extracted with 99% methanol. The samples were vortexed for 30 seconds and centrifuged at 20°C for 10 minutes at 5000 rpm. Then the supernatant was removed and poured into a new separate tube. Again, methanol was added to the remaining leaf material. After vortexing and centrifugation, then, again supernatant was removed and added to the previous solution. After this, the entire supernatant solution was placed in a vacuum device for 30 minutes to concentrate the extract. Then this extract was used to measure total phenols.

The most serious objection to the content of the paper is the lack of a summary in the form of conclusions of the study. This section should be added to the paper.

Response: We have included a Conclusion section.

Reviewer 2 Report (Previous Reviewer 3)

Comments and Suggestions for Authors

Though the authors have put a lot of effort to improve their manuscript well, but I have some suggestions and recommendations which authors should follow before proceeding to the next step. The manuscript requires careful revision.

1) Line 34: Change ‘compared to’ into ‘compared with’.

2) Line 88: the sentence is confusing.

3) Line 93: ‘Yadollahi et al 20108’ is wrongly cited. Yadollahi et al 2011 and Yadollahi et al 2010 are cited in the text, but only one reference is listed.

4) Line 91: Revise the sentence ‘Also Root (dry weight)/(leaf area)’.

5) Line 135: Change ‘29,88’ into ‘29.88’.

6) Figure 1 and Figure 2: ‘-1’ should be superscript.

7) Line 362: Change ‘character-istics’ into ‘characteristics’.

8) Table 1: ‘-1’ should be superscript.

9) Line 466-483: the number in ‘H2O2’ should be subscript.

10) Line 481: the number in ‘NaN3‘ should be subscript.

11) Reference 67 and 68 are the same.

Comments on the Quality of English Language

 The English must be improved to more clearly express the research.

Author Response

Dear Editor and Reviewers,

The author would like to gratefully acknowledge and sincerely thank the reviewers for their valuable comments. We have responded to all comments and have revised the paper in light of them. Detail of our responses to each comment are shown below. Comments, when requested, are shown in red.

We would like to comment that a complete English revision has been performed by the MDPI English service. In addition, new figures were created and all Annex were removed.

Reviewer 2)

Though the authors have put a lot of effort to improve their manuscript well, but I have some suggestions and recommendations which authors should follow before proceeding to the next step. The manuscript requires careful revision.

1) Line 34: Change ‘compared to’ into ‘compared with’.

Response: We have modified the text  

2) Line 88: the sentence is confusing.

            Response: We have modified the text

3) Line 93: ‘Yadollahi et al 20108’ is wrongly cited. Yadollahi et al 2011 and Yadollahi et al 2010 are cited in the text, but only one reference is listed.

            Response: We have modified the text. The correct is Yadollahi et al 2010

4) Line 91: Revise the sentence ‘Also Root (dry weight)/(leaf area)’.

            Response: We have modified the text

5) Line 135: Change ‘29,88’ into ‘29.88’.

Response: We have modified the text

6) Figure 1 and Figure 2: ‘-1’ should be superscript.

            Response: We have improved the figures.

7) Line 362: Change ‘character-istics’ into ‘characteristics’.

            Response: We have modified the text

8) Table 1: ‘-1’ should be superscript.

            Response: We have modified the table

9) Line 466-483: the number in ‘H2O2’ should be subscript.

            Response: We have modified the text

10) Line 481: the number in ‘NaN3‘ should be subscript.

            Response: We have modified the text.

11) Reference 67 and 68 are the same.

            Response: We have removed one reference

Reviewer 3 Report (New Reviewer)

Comments and Suggestions for Authors

The identification and selection of drought-tolerant genotypes contribute to breeding more resilient almond cultivars. This manuscript systematically investigates the physiological and biochemical responses of 14 distinct almond genotypes under water deficit conditions. The findings indicate that "Shahrood 8", "Garnem", and "Shahrood 12" demonstrate superior drought stress adaptation. While the manuscript presents comprehensive research with accurate analysis and thorough discussion, several modifications are recommended:

1. Include statistical significance annotations and error bars in the figures.

2. Provide pictures of plant growth.

3. Improve Figure 1H.

4. Relocate figure captions below the corresponding figures.

5. There are only two images in a manuscript that are too few. It is recommended to split Figure 1 according to indicator categories.

6. Revise total phenolic content units to mg GAE/g FW.

7. Line 231 should be Supplementary Figure A4 and Figure A3.

8. It is recommended to include some appendix figures into the main manuscript.

9. Review subscript formatting in lines 466, 467, 470, and 483.

Author Response

Dear Editor and Reviewers,

The author would like to gratefully acknowledge and sincerely thank the reviewers for their valuable comments. We have responded to all comments and have revised the paper in light of them. Detail of our responses to each comment are shown below. Comments, when requested, are shown in red.

We would like to comment that a complete English revision has been performed by the MDPI English service. In addition, new figures were created and all Annex were removed.

Reviewer 1)
The identification and selection of drought-tolerant genotypes contribute to breeding more resilient almond cultivars. This manuscript systematically investigates the physiological and biochemical responses of 14 distinct almond genotypes under water deficit conditions. The findings indicate that "Shahrood 8", "Garnem", and "Shahrood 12" demonstrate superior drought stress adaptation. While the manuscript presents comprehensive research with accurate analysis and thorough discussion, several modifications are recommended:

Include statistical significance annotations and error bars in the figures.
            Response: New Figures have been generated with all this information.

Provide pictures of plant growth.
            Response: Thank you very much for your suggestion,  but we regret inform that we cannot provide pictures of the plant growth. An important issue happened with the computer of the student (first author), it was stolen and all the information was lost three months ago. To demonstrate the reality of the experiment and to confirm the experimental trials, we would like to indicate that a previous paper only with morphological traits (crown length, leaf area, sub-branch length, num. of internode of sub-branch, and macro and micro elements in leaves) in the same trees, same RCBD, was published in an Iranian Journal of Horticultural Science, (I hope you can understand the whole issue).

Safavi, E., Yadegari, M., Mousavi, S.A., & Haghighati, B. (2023). Investigation the different levels of drought stress on almond cultivars. Journal of Horticultural Science, 37(2), 523-540. (In Persian with English abstract). https://doi.org/10.22067/jhs.2022.77478.1184

Improve Figure 1H.
Response: All figures have been improved

Relocate figure captions below the corresponding figures.
Response: Captions have been located below the corresponding figures

There are only two images in a manuscript that are too few. It is recommended to split Figure 1 according to indicator categories.
Response: A new set of 5 Figures are now in the manuscript.

Revise total phenolic content units to mg GAE/g FW.
            Response: We have revised the units

Line 231 should be Supplementary Figure A4 and Figure A3.
            Response: This information has been updated with the new figures.

It is recommended to include some appendix figures into the main manuscript.
Response: A new set of 6 Figures are now in the manuscript.

Review subscript formatting in lines 466, 467, 470, and 483.
Response: The subscript has been updated in each line

Reviewer 4 Report (New Reviewer)

Comments and Suggestions for Authors

Comments to the Author

Overall comments

This study analyzed the physiological and biochemical adaptations of Almond (Prunus dulcis) cultivars to drought stress in semi-arid regions of Iran. Although, previous studies have explored similar drought tolerant traits in almonds but this research still holds significant implications for agricultural sustainability of almond cultivation in Iran. The experiment is well-designed and data itself is worthy of publication however, writing needs substantial improvement. I have provided the specific comments to improve the paper title and abstract. The introduction and discussion section lacks a clear flow and contains lot of irrelevant information. Moreover, many areas suffer from grammatical errors. I also felt that discussion was overly simplistic (largely repeating results) and did not significantly enhance our knowledge. I recommend expanding the discussion to explore the underlying mechanisms and broader implications of the findings. Finally, I recommend a thorough revision to enhance clarity, coherence and readability. I also have provided some specific suggestions below, which will greatly improve the paper but I think it will require an extensive overhaul/rewrite to get there.  

Specific comments

1.       Line 2. I suggest changing the paper title to Physiological and biochemical adaptations of Almond (Prunus dulcis) cultivars to drought stress in semi-arid regions of Iran

2. Line 18-20: Consider revising “Identifying and selecting drought-tolerant almond cultivars is crucial for developing more resilient cultivars, especially in regions prone to water scarcity affected by global climate change” to improve the sentence flow.

3.       Line 20: Suggest starting the sentence form “In this study, the physiological and biochemical responses ……………….”

4.       Line 24-27:  Consider deleting these two sentences. A first sentence is repetitive with L.20 as already described before the physiological and biochemical traits of almond were evaluated whereas 2nd sentence is unnecessary here and can be omitted for better clarity. Moreover, in results section you say the different in these traits in first and second year whereas in abstract, you state that  they exhibited a similar trend in both years. You need to clarify, which statement is correct.

5.       Line 27: Suggest starting this sentence from “The results showed that chlorophyll a and b content …………………”       

6.       Line 32: Consider deleting the phrase “According to the results” and start the sentence with the cultivars …………………………………”

7.       The last sentence of the abstract should explicitly convey the significance of this study’s findings e.g. these results provide valuable insights ………………………“ but its lacking in your abstract section.

8.       Line 38: First paragraph of introduction lacks clarity and coherence with lot of grammatical errors. It’s recommended to revise whole paragraph to improve the sentences flow effectively.      

9.       Line 40-45: Your study focuses on semi-arid regions of Iran while discussion about effects of drought on Mediterranean basin and California seems strange here.

10.   Line 50: This paragraph appears to be highly irrelevant. I suggest revising it to focus on the various physiological and biochemical adaptations employed by plants to cope with drought stress and how identifying these mechanisms in response to drought will significantly enhance our knowledge.  

11.   Line 67: you used the phrase “in this sense” many times throughout the manuscript that seems unnecessary. Please consider deleting it and start this paragraph from “To improve the drought resistance in almonds…………………….”

12.   Line 75-76: The phrase ‘not affected by drought’ and ‘well maintained under drought’ convey similar meanings. I recommend selecting one of these expressions to avoid redundancy and improve the clarity of sentence.  

13.   Line 77: consider deleting the phrase “In all these studies” and start the sentence with “Various drought related traits………...”. Moreover, it’s strongly recommended to merge this paragraph with previous one that would improve the flow.  

14.   Line 88: I recommend rewriting of this section. A lot of sentences are overly long and complex with lot of grammatical mistakes. I suggest them to breaking long sentences into shorter and clearer sentences to improve the readability.  

15.   Line 102: This section need to be rephrased thoroughly for clarity.

16.   Line 116: Figure captions for Fig 1A (also other supplementary figures) are missing. Its highly recommended figure captions for each supplementary figure to help the readers understand the content e.g. in Fig 1A, the data for the first and the second year are unclear, and the y-axis labels related to drought are not well-defined. I recommend clarifying of these points for better understanding.

17.   Line 147: The caption for Figure 1 should be placed below the figure for consistency.

18.   Line 156: Which figures compare the chlorophyll content of first and second year? Clarify it.

19.   Line 204: rephrase the sentence for clarity.

20.   Line 206: It would be better to discuss the potential factors for the variation of leaf physiological and biochemical traits in first and second year.

21.   Line 250: I suggest starting the discussion about importance of studying physiological or biochemical adaptations to drought. Perhaps you could mention how semi-arid regions, such as those in Iran, are particularly vulnerable to increasing aridity and water scarcity due to climate change, highlighting the relevance of your research.    

22.   Line 276: Most of these parameters not studied in this work, I recommend removing this part or revising it to focus on the parameters relevant to this study.

23.   Line 294-299: Consider revising this section for clarity.

24.   Line 358: Consider revising it to “The experiment was conducted at the Agricultural and Natural Resources Research Center in Shahrekord, Iran (53° 17' N, 56° 55' E; altitude 2073 m), using a split-plot design based on randomized complete block design (RCBD) during the 2020 and 2021 growing seasons”. You should specify whether the experiment was conducted under natural field conditions or in a controlled environment, such as greenhouse.

25.   Line 489: This information should include in the experimental design section. The statistical analysis section should focus on how the data was analyzed and statistical methods used.  

26.   Figure 1 and 2: You mentioned drought stress treatments as 10, 30, 50, 70; however, you should specify if these values are in percentages and either it states the drought stress or field capacity in percentages.

27.   The figures provided (especially supplementary figures) are in very bad quality and difficult to see. I strongly recommend replacing them with high-resolution (300d dpi) to ensure better visual presentation. 

Comments on the Quality of English Language

The manuscript contains numerous grammatical errors. A thorough language edit by a professional editor or a native English speaker is strongly recommended to address these issues and improve the overall clarity and quality of the text.

Author Response

Dear Editor and Reviewers,

The author would like to gratefully acknowledge and sincerely thank the reviewers for their valuable comments. We have responded to all comments and have revised the paper in light of them. Detail of our responses to each comment are shown below. Comments, when requested, are shown in red.

We would like to comment that a complete English revision has been performed by the MDPI English service. In addition, new figures were created and all Annex were removed.

Reviewer 4)

Comments to the Author

Overall comments

This study analyzed the physiological and biochemical adaptations of Almond (Prunus dulcis) cultivars to drought stress in semi-arid regions of Iran. Although, previous studies have explored similar drought tolerant traits in almonds but this research still holds significant implications for agricultural sustainability of almond cultivation in Iran. The experiment is well-designed and data itself is worthy of publication however, writing needs substantial improvement. I have provided the specific comments to improve the paper title and abstract. The introduction and discussion section lacks a clear flow and contains lot of irrelevant information. Moreover, many areas suffer from grammatical errors. I also felt that discussion was overly simplistic (largely repeating results) and did not significantly enhance our knowledge. I recommend expanding the discussion to explore the underlying mechanisms and broader implications of the findings. Finally, I recommend a thorough revision to enhance clarity, coherence and readability. I also have provided some specific suggestions below, which will greatly improve the paper but I think it will require an extensive overhaul/rewrite to get there. 

Specific comments

  1. Line 2. I suggest changing the paper title to Physiological and biochemical adaptations of Almond (Prunus dulcis) cultivars to drought stress in semi-arid regions of Iran

Response: We have improved the text accordingly.

  1. Line 18-20: Consider revising “Identifying and selecting drought-tolerant almond cultivars is crucial for developing more resilient cultivars, especially in regions prone to water scarcity affected by global climate change” to improve the sentence flow.

Response: A complete English revision has been performed by the MDPI English service.

  1. Line 20: Suggest starting the sentence form “In this study, the physiological and biochemical responses ……………….”

                Response: We have improved the text accordingly.

  1. Line 24-27: Consider deleting these two sentences. A first sentence is repetitive with L.20 as already described before the physiological and biochemical traits of almond were evaluated whereas 2nd sentence is unnecessary here and can be omitted for better clarity. Moreover, in results section you say the different in these traits in first and second year whereas in abstract, you state that  they exhibited a similar trend in both years. You need to clarify, which statement is correct.

Response: We have improved the text accordingly.

  1. Line 27: Suggest starting this sentence from “The results showed that chlorophyll a and b content …………………”

Response: We have improved the text accordingly.

  1. Line 32: Consider deleting the phrase “According to the results” and start the sentence with the cultivars …………………………………”

Response: We have improved the text accordingly.

  1. The last sentence of the abstract should explicitly convey the significance of this study’s findings e.g. these results provide valuable insights ………………………“ but its lacking in your abstract section.

Response: We have improved the text accordingly.

  1. Line 38: First paragraph of introduction lacks clarity and coherence with lot of grammatical errors. It’s recommended to revise whole paragraph to improve the sentences flow effectively.

Response: We have improved the text accordingly.

  1. Line 40-45: Your study focuses on semi-arid regions of Iran while discussion about effects of drought on Mediterranean basin and California seems strange here.

Response: We have improved the text accordingly.

  1. Line 50: This paragraph appears to be highly irrelevant. I suggest revising it to focus on the various physiological and biochemical adaptations employed by plants to cope with drought stress and how identifying these mechanisms in response to drought will significantly enhance our knowledge.

Response: We have improved the text accordingly.

  1. Line 67: you used the phrase “in this sense” many times throughout the manuscript that seems unnecessary. Please consider deleting it and start this paragraph from “To improve the drought resistance in almonds…………………….”

We have improved the text accordingly.

  1. Line 75-76: The phrase ‘not affected by drought’ and ‘well maintained under drought’ convey similar meanings. I recommend selecting one of these expressions to avoid redundancy and improve the clarity of sentence.

We have improved the text accordingly.

  1. Line 77: consider deleting the phrase “In all these studies” and start the sentence with “Various drought related traits………...”. Moreover, it’s strongly recommended to merge this paragraph with previous one that would improve the flow.

                Response: We have improved the text accordingly.

  1. Line 88: I recommend rewriting of this section. A lot of sentences are overly long and complex with lot of grammatical mistakes. I suggest them to breaking long sentences into shorter and clearer sentences to improve the readability.

Response: We have improved the text accordingly.

  1. Line 102: This section need to be rephrased thoroughly for clarity.

Response: We have improved the text accordingly.

  1. Line 116: Figure captions for Fig 1A (also other supplementary figures) are missing. Its highly recommended figure captions for each supplementary figure to help the readers understand the content e.g. in Fig 1A, the data for the first and the second year are unclear, and the y-axis labels related to drought are not well-defined. I recommend clarifying of these points for better understanding.

Response: We have improved ALL THE FIGURES accordingly.

  1. Line 147: The caption for Figure 1 should be placed below the figure for consistency.

Response: We have improved ALL THE FIGURES accordingly

  1. Line 156: Which figures compare the chlorophyll content of first and second year? Clarify it.

Response: We have improved ALL THE FIGURES accordingly. It is compare with the anova results in the supplemental file1.

  1. Line 204: rephrase the sentence for clarity.

Response: We have improved the text accordingly.

  1. Line 206: It would be better to discuss the potential factors for the variation of leaf physiological and biochemical traits in first and second year.

Response: We have improved the text accordingly.

  1. Line 250: I suggest starting the discussion about importance of studying physiological or biochemical adaptations to drought. Perhaps you could mention how semi-arid regions, such as those in Iran, are particularly vulnerable to increasing aridity and water scarcity due to climate change, highlighting the relevance of your research.

Response: We have improved the text accordingly.

  1. Line 276: Most of these parameters not studied in this work, I recommend removing this part or revising it to focus on the parameters relevant to this study.

Response: We have improved the text accordingly.

  1. Line 294-299: Consider revising this section for clarity.

Response: We have improved the text accordingly.

  1. Line 358: Consider revising it to “The experiment was conducted at the Agricultural and Natural Resources Research Center in Shahrekord, Iran (53° 17' N, 56° 55' E; altitude 2073 m), using a split-plot design based on randomized complete block design (RCBD) during the 2020 and 2021 growing seasons”. You should specify whether the experiment was conducted under natural field conditions or in a controlled environment, such as greenhouse.

Response: We have improved the text accordingly.

  1. Line 489: This information should include in the experimental design section. The statistical analysis section should focus on how the data was analyzed and statistical methods used.

Response: We have improved the text accordingly.

  1. Figure 1 and 2: You mentioned drought stress treatments as 10, 30, 50, 70; however, you should specify if these values are in percentages and either it states the drought stress or field capacity in percentages.

Response: We have improved the FIGURES accordingly.

  1. The figures provided (especially supplementary figures) are in very bad quality and difficult to see. I strongly recommend replacing them with high-resolution (300d dpi) to ensure better visual presentation.

Response: We have REMOVED THE FIGURES IN ANEX 1 

Round 2

Reviewer 1 Report (Previous Reviewer 2)

Comments and Suggestions for Authors

Authors improved the manuscript according my suggestions.

Author Response

The authors would like to gratefully acknowledge and sincerely thank the reviewer for their valuable comments. 

Reviewer 2 Report (Previous Reviewer 3)

Comments and Suggestions for Authors

The authors have addressed most of my previous comments. However, its current presentation and style as well as English language expression should be much improved since there were demerits as listed following:

  1.  Line 98: Change “(Martínez-García et al., 2020”into “(Martínez-García et al., 2020)”.
  2.  Line 100: Change “ in compared to”into “in comparison to”.
  3.  Figure 1: the unit for chlorophyll a (Cha) and chlorophyll b (Chb) contents in the figure is µmol g-1 FW. However, the unit is mg g-1 FW in the text and figure title.
  4.  Line 152: Change “compared to” into “compared with”.
  5.  Figure 4: the unit for total phenol content in the figure is GAE/g FW . However, it is mg g-1 FW in the text
  6.  Line 497: “µmol MDA per g of leaf FW” is confusing.
Comments on the Quality of English Language

 English language expression should be much improved. 

Author Response

The authors would like to gratefully acknowledge and sincerely thank the reviewer for their valuable comments. 

The manuscript was reviewed thoroughly by MPDI English service again, finding very few issues with the expression of English; few minor changes were made to slightly improve the expression of the manuscript.

  1.  Line 98: Change “(Martínez-García et al., 2020”into “(Martínez-García et al., 2020)”.
      1. Response: The line has been changed.
  2.  Line 100: Change “ in compared to”into “in comparison to”.
      1. Response:The line has been changed.
  3.  Figure 1: the unit for chlorophyll a (Cha) and chlorophyll b (Chb) contents in the figure is µmol g-1 FW. However, the unit is mg g-1 FW in the text and figure title.
      1. Response:The figures and text have been updated.
  4.  Line 152: Change “compared to” into “compared with”.
      1. Response:The line has been changed.
  5.  Figure 4: the unit for total phenol content in the figure is GAE/g FW . However, it is mg g-1 FW in the text
      1. Response: Thes figures and tex has been changed accordingly
  6.  Line 497: “µmol MDA per g of leaf FW” is confusing.
      1. Response: The units have been changed to  µmol g−1 FW to avoid confusion.

Reviewer 3 Report (New Reviewer)

Comments and Suggestions for Authors The author has responded to all comments and made revisions to the manuscript. There is a formatting issue with Figure 2-5 in the current manuscript, which may be caused by the PDF version.

Author Response

The authors would like to gratefully acknowledge and sincerely thank the reviewer for their valuable comments. All figures have been adjusted to the manuscript.

Reviewer 4 Report (New Reviewer)

Comments and Suggestions for Authors

I appreciate the authors’ thorough revisions and thoughtful responses to my previous comments. The manuscript has significantly improved in clarity. All my concerns have been adequately addressed, and I am satisfied with the current version. I recommend the manuscript for acceptance in its present form.

Author Response

The authors would like to gratefully acknowledge and sincerely thank the reviewer for their valuable comments. 

This manuscript is a resubmission of an earlier submission. The following is a list of the peer review reports and author responses from that submission.

Round 1

Reviewer 1 Report

Comments and Suggestions for Authors

Much attention is paid to how the data were processed and how the statistical analysis was done, but in fact the experimental setup is not clearly described. It is not clear how the drought was applied and how the plants were protected from the climatic factors. The equipment information is missing, eg spectrophotometer, trademark, model, etc. 

Regarding the results, the RWC should be given first, and only after that the results for the MDA, etc. Table 3 presents data for MDA, RWC and EL and although it is a table, it is relatively clear. But Fig. 1A and 2A do not carry any information at all. They present averaged data from the varieties, but the idea is to consider the data by variety. Fig. 1B and 2B is also not informative. Please, think about another way of presenting the data that includes the different % water deficits. 

The description of the tables and graphs is too short.

The Discussion should be structured to show and seek explanations for differences between the varieties. 

The first two sentences in Conclusions describe the purpose of the work, their place is not there. 

Three varieties are identified as the most resistant, but are not mentioned anywhere in the discussion. On what basis are they defined as the most sustainable?

There is a discussion in the Conclusions section that should be placed in the Discussion section.

Comments on the Quality of English Language

Minor editing of English language is required.

Reviewer 2 Report

Comments and Suggestions for Authors

In the publication, the authors presented a detailed study of large groups of almond trees subjected to long-term drought stress. Water deficiency accounted for a different percentage for the various groups studied. The content of enzymes, carbohydrates, polyphenols and antioxidant capacity were then examined in relation to the different percentages of water deficit. The research was conducted over two growing seasons.

The paper is very well written. The authors could have developed the conclusions subsection to highlight the most important findings of the research, which could contribute to further almond breeding.

Reviewer 3 Report

Comments and Suggestions for Authors

The manuscript entitled “The variation of physiological and biochemical traits of almond (Prunus dulcis [Mill.] D.A. Webb) in response to drought stress under semi-arid condition of Iran” has been reviewed and found that the manuscript contained all the information which will be useful for the research community. I have some major suggestions to make this submission more attractive.

Q1. The language needs to be substantially improved. Native language speaker/professional proofreading service is mandatory.

• The keywords should not be included in the title.

• All genes' names should appear in italics. Please check the entire text and correct them.

Q2. The title is misleading. It should revised in more meaningful way like “Exploring Physiological and Biochemical Responses of Almond (Prunus dulcis) to Drought Stress in Semi-Arid Iran” or “Unraveling Almond (Prunus dulcis) Adaptations to Drought Stress in Semi-Arid Iran: Insights into Physiological and Biochemical Changes”

Q3. Abstract contained very general information. I suggest to rewrite the abstract in more meaningful way. Abstract should contain the key results and make it good for researcher audience.

Q4. The introduction is not well written as it does not provide a comprehensive background for the study presented in this manuscript. For example, line 35-36, which kind of horticulture crops are majorly facing the drought and what kind of impact on its yield and quality.

Q5. Materials and methods

a. There are no details about the medium/soil used for plant growing.

b. On which basis, 13 commercial almond cultivars were selected? Highlight the importance of the cultivar used in the manuscript.

c. What about the agronomic practices, did not find any information?

Q6. Results

a. In general the results are not clearly described. There are no relations between subsequent sections-each section is a separate piece of results not connected to other results presented in this manuscript. The description of the results is rather shallow for example “the notably, the EL and MDA levels measured in the second year were found to be lower than those observed in the first year” should include values not only a generic statement. The text is also very chaotic

b.  “The variance analysis results indicated that the proline content of almond leaves was significantly influenced by the year, drought stress, cultivar, and the interaction effect of drought × cultivar.” – not clear

c.  The mean squares of year, drought, and cultivar were significant factors affecting peroxidase activity – is it so basic and obvious information and here it seems that the Authors added lot of generic information in results.

d. Lines 328-330: “Drought stress induced a significant increase in SOD activity in all almond cultivars, with the highest activity observed under severe drought stress. Under non-stress conditions, "Shahrood 12" and "Saba" cultivars exhibited the highest and lowest SOD activity, respectively”. Very generic information

e. Lines 324-326: The recorded GPX activity in the second studied year was higher than in the first year. Under non-stress conditions, cultivars "Rabie”, "Shahrood 21”, "Shahrood 13”, and "Mamaei" had the lowest GPX activity, while "Garnem”, "Shahrood 8”, and "Shahrood 12" cultivars exhibited the highest activity? Please rewrite along with results instead of general information.

f. Strongly suggest to rewrite the result section in meaningful way. Add real-time results instead of giving general information.

Q7. Discussion

· Amixture of repetition of the results and some basic information from the literature. The discussion should be revised.

· In between the sentences there is a lack of connectivity such as in lines 348 to 350. Should all some back ground and previously research regarding horticulture crops especially fruit crops.

· Line 404-406: Need to be precise. How the statements are connected to the present context. Author should make contact with previous ongoing talk. No relation was found between sentences.

· It seems to be more speculative. Authors need to strengthen the justification of their findings.

Q8. Conclusions

The summary of results. Please provide proper conclusions based on the results presented in this manuscript.

Q9. Figure captions:

a. Fig. 2: should be more informative. 

Comments on the Quality of English Language

The language needs to be substantially improved. Native language speaker/professional proofreading service is mandatory.